# Differential Cerebral Gustatory Responses to Sucrose, Aspartame, and Stevia Using Gustatory Evoked Potentials in Humans

**DOI:** 10.3390/nu12020322

**Published:** 2020-01-27

**Authors:** Thomas Mouillot, Anaïs Parise, Camille Greco, Sophie Barthet, Marie-Claude Brindisi, Luc Penicaud, Corinne Leloup, Laurent Brondel, Agnès Jacquin-Piques

**Affiliations:** 1Centre des Sciences du goût et de l’Alimentation, AgroSup Dijon, CNRS, INRA, Université Bourgogne Franche-Comté, F-21000 Dijon, France; thomas.mouillot@chu-dijon.fr (T.M.); an.parise@free.fr (A.P.); greco.camille004@gmail.com (C.G.); Sophie.Barthet@inra.fr (S.B.); marie-claude.brindisi@chu-dijon.fr (M.-C.B.); luc.penicaud@inserm.fr (L.P.); corinne.leloup@u-bourgogne.fr (C.L.); Laurent.Brondel@u-bourgogne.fr (L.B.); 2Department of Hepatology and Gastroenterology, 14, CHU Dijon Bourgogne, Rue Paul Gaffarel, F-21000 Dijon, France; 3Department of Endocrinology and Nutrition, 14, CHU Dijon Bourgogne, Rue Paul Gaffarel, F-21000 Dijon, France; 4Department of Clinical Neurophysiology, 14, CHU Dijon Bourgogne, Rue Paul Gaffarel, F-21000 Dijon, France

**Keywords:** Gustatory evoked potentials, sucrose, sweeteners, aspartame, Stevia

## Abstract

Aspartame and Stevia are widely substituted for sugar. Little is known about cerebral activation in response to low-caloric sweeteners in comparison with high-caloric sugar, whereas these molecules lead to different metabolic effects. We aimed to compare gustatory evoked potentials (GEPs) obtained in response to sucrose solution in young, healthy subjects, with GEPs obtained in response to aspartame and Stevia. Twenty healthy volunteers were randomly stimulated with three solutions of similar intensities of sweetness: Sucrose 10 g/100 mL of water, aspartame 0.05 g/100 mL, and Stevia 0.03 g/100 mL. GEPs were recorded with EEG (Electroencephalogram) electrodes. Hedonic values of each solution were evaluated using the visual analog scale (VAS). The main result was that P1 latencies of GEPs were significantly shorter when subjects were stimulated by the sucrose solution than when they were stimulated by either the aspartame or the Stevia one. P1 latencies were also significantly shorter when subjects were stimulated by the aspartame solution than the Stevia one. No significant correlation was noted between GEP parameters and hedonic values marked by VAS. Although sucrose, aspartame, and Stevia lead to the same taste perception, cerebral activation by these three sweet solutions are different according to GEPs recording. Besides differences of taste receptors and cerebral areas activated by these substances, neural plasticity, and change in synaptic connections related to sweet innate preference and sweet conditioning, could be the best hypothesis to explain the differences in cerebral gustatory processing after sucrose and sweeteners activation.

## 1. Introduction

Artificial and natural sweeteners are widely substituted for sugar and are present in many products, especially in sugar-free products. Their use is thought to reduce sugar consumption and significantly decrease caloric intake while maintaining the palatability of food and soft drinks. They are also becoming more important in terms of dietary restrictions for diabetes [1]. Among the artificial sweeteners, aspartame is commonly used throughout the food and drinks industry and is approximately 200 times sweeter than sucrose [2]. Steviol glycosides are natural sweeteners extracted from a native shrub of Brazil and Paraguay, Stevia rebaudiana Bertoni (Stevia). The sweetener mixture mainly consists of stevioside and rebaudioside-A (Steviol glycosides) [3,4]. Stevioside is reported to be 250–300 times and rebaudioside-A 300–400 times sweetener than sucrose [5,6]. 

The use of sweeteners is increasingly controversial in the literature, especially for aspartame [7,8]. Among the disorders caused by the consumption of aspartame, neurological and behavioral disorders have been reported, including headache, insomnia, and seizures. These disorders may be due to changes in regional brain concentrations of catecholamines induced by the metabolites of aspartame [1]. The steviol glycosides from Stevia represent a very interesting alternative. However, the physiological mechanisms of action of these glycosides are still unknown, particularly concerning cerebral activation induced by this natural sweetener [6].

The sweet taste of both sugar and sweeteners is mediated by metabotropic receptors T1R2 and T1R3 [9]. Sugar then activates the primary gustatory cortex (insula and opercular cortex) and the reward system (amygdala, orbitofrontal, anterior cingulate, and prefrontal cortices) [10,11]. In rats, it has been demonstrated that Stevia consumption leads to modification in the activation of the mesolimbic dopamine reward system [12]. However, no study has compared cerebral activation using fMRI between sugar and natural sweeteners such as Stevia in Humans. Only one study observed that Stevia could decrease appetite sensation, but this study was conducted with subjective tests [13]. Moreover, little is known about the peripheral and central gustatory pathways that are activated by artificial sweeteners, or about the differences that might exist in the activation of central feeding and reward systems between sugar and sweeteners. It has been observed, using functional magnetic resonance imaging (fMRI), that sucrose or glucose might be able to more strongly recruit the reward-related brain regions (striatum, anterior cingulate, and prefrontal cortices) than artificial sweeteners (sucralose in Frank et al. [14], and saccharin in Chambers et al. [15]), despite the inability of subjects to consciously distinguish the tastes. To our knowledge, only one previous study observed discrepancies in brain activation after stimulation by different sweet solutions (sucrose and aspartame) using electroencephalography (EEG) and Gustatory Evoked Potentials (GEPs) methods [16]. However, no study using GEPs was conducted with gustatory stimulation by natural sweeteners like Stevia. 

GEPs are a non-invasive method with high time resolution to study the whole gustatory pathway [17]. GEPs have already been obtained in response to sugar solutions such as sucrose [17,18,19], demonstrating the activation of the gustatory pathway from the tongue receptors to the taste cortex by sucrose solutions. GEP was defined by 3 peaks: P1, N1, and P2. The P1 peak corresponds to the beginning of the GEP. P1 and N1 peaks are described as the sensory cerebral response [17]; the P1N1 amplitude corresponds to the intensity of the cerebral activation. The P2 peak is described as the subjective interpretation of the gustative stimulus by the subject (cognitive response). 

In this study, we hypothesized that GEPs could be obtained in response to tongue stimulation by natural-like artificial sweeteners because they are able to bind metabotropic receptors like sugar. We also hypothesized that cortical activation could be less intense with sweeteners than with sugar, leading to record GEPs that might have longer latencies and smaller amplitudes in response to sweeteners than sugar. In fact, previous studies observed lower activation of the reward-related brain regions by sweeteners than sucrose [14,15]. To test these hypotheses, we aimed to compare GEPs obtained in response to sucrose solution in young, healthy subjects, with GEPs obtained in response to aspartame, an artificial sweetener, and to Stevia, a natural sweetener, with the same experimental design.

## 2. Materials and methods

### 2.1. Subjects

Twenty healthy subjects, 10 men and 10 women, were enrolled in this study. The mean age and BMI (body mass index) were 22 ± 2 years old (range: 19 to 27 years old) and 23 ± 3 kg/m^2^ (range: 17 to 27 kg/m^2^), respectively. The mean age was 21 ± 2 years old for the women and 22 ± 3 for the men. The mean BMI was 22 ± 3 kg/m^2^ for women and 24 ± 2 kg/m^2^ for men. 

All of the subjects were non-smokers. None of the subjects had oral, dental, or neurological disorders or specific medical histories. Subjects who were currently undergoing medical treatment and obese subjects (BMI > 30 kg/m^2^) were excluded. All the participants practiced regular physical activity (about 30 min, 3 times per week). They did not have a special diet (vegan, vegetarian, etc.), and their meal was balanced. They rarely consumed sweeteners or prepared foods that contain sweeteners (<1 time per month). The participants drank about 1 to 3 coffees per day. They drank alcohol occasionally (1–3 glasses per week), without abnormality according to WHO (World Health Organization) guidelines.

### 2.2. Ethical Approval

The subjects were informed about the nature and aims of the experiments and provided with informed consent. The study was approved by the Regional Ethics Committee of Burgundy, France, in accordance with the latest revision of the Declaration of Helsinki and European Law (ISO EN 14155).

### 2.3. Experimental Design

The taste delivery system was detailed in previous studies [18,19]. Control and sweet solutions were driven through the system by compressed air (controlled through a manometer). Two parallel silicone tubes were used; one for the control solution (water) and the other for the sweet solution. Switching between the control and sweet solutions was performed by two electromagnetic valves controlled by an electronic device. This electronic device (stimulator) sent a signal (the trigger) to the computer software (SystemPLUS EVOLUTION, 2007 Micromed S.p.A) when the sweet solution was administered (with 1 ms precision), resulting in a precise time recording of the GEPs. Participants put the 2 parallel tubes, 1 for the sweet solution and 1 for water, on the middle of their tongue in their mouth. Air was purged from the taste delivery system to avoid delaying stimulus presentation. 

The ends of the semi-rigid tubes (silicone tubing, P/N 10025-02S, Bio-Chem valve) were placed 1.5 ± 0.5 cm from the dental arch on the midline of the tongue (same distance for each subject). Due to their rigidity, the tubes could not deviate from this position. Solutions were delivered to the tongue through a hole at the end of each tube. A sweet solution was intermittently delivered through the first tube (flow rate = 200 mL/h). During the period without the sweet solution, water was continuously delivered through the second tube (flow = 100 mL/h) to minimize the likelihood that the subjects would feel different sensations from the injections from the two tubes. A 200 mL/h flow rate was chosen for the sweet solution because it allowed uniform stimulation of a large lingual surface (21 to 24 cm^2^, tested just after stimulation of the tongue using a methylene blue solution in a preliminary study), and thus that the same oral receptive fields were activated in each subject. A 100 mL/h flow rate was chosen for water because it did not induce somatosensory differences compared to the flow rate of 200 mL/h, and this flow rate did not oblige participants to swallow frequently, which could generate artifacts in the GEP recording. This was tested in a preliminary study and verified in the control recordings. 

#### 2.3.1. Sweet Solutions

The sweet stimuli consisted of sucrose (Cooper—10 g/100 mL of water), aspartame (Cooper—0.05 g/100 mL of water), and Stevia (“Stevia ext 97% rebaudioside A 50 G Cooper”—0.03 g/100 mL of water) solutions, diluted in Evian water. The solutions were prepared just before GEP recording. Evian water, which was almost deionized, was used as the control solution. Stimulation with Evian water alone did not induce GEP [19]. 

Concentrations of the solutions were chosen according to the sweetening power of the substances, tested in a preliminary questionnaire, using a 10 cm visual analog scale (VAS) anchored by “not at all” and “extremely” at its extremities. The subjects had to respond to the following question: “How intense was the sweet solution?” The solutions containing 10 g of sucrose, 0.05 g of aspartame, and 0.03 g of Stevia were perceived with similar intensities of sweetness and without a detectable aftertaste, which was tested before the beginning of the present study in 20 different subjects.

#### 2.3.2. Sweet Solution Stimulations

Subjects were investigated in 3 double-blind sessions separated by an interval of at least 1 day. Each session corresponded to a specific stimulus quality, which was randomly assigned. All of the sessions were conducted at the same time of day for each subject, 2–4 h after lunch. The subjects were asked not to eat or drink anything except water during the time between lunch and the GEP recording. One session lasted approximately 40 min: 20 min to prepare for the GEP recording and 20 min for the GEP recording itself. In each session, the sweet stimulus was presented for 1 second 20 times. Each stimulus was separated by a 1 minute interval of water solution.

During the GEP recordings, the subjects listened to quiet music of their preference through their headphones to mask the switching clicks of the electromagnetic valves. Moreover, the valves were kept in a sound-attenuating box. No evoked potentials were recorded in our experiment in response to quiet music (checked with control GEP recordings) [19]. The subjects also had to close their eyes to avoid light stimulation. 

After the GEP recordings performed with the sweet solutions, the subjects were asked to rate the hedonic value of each solution using a 10 m visual analog scale (VAS) anchored by “not at all” and “extremely” at its extremities. They had to respond to the following question: “How palatable was the taste solution?”. The subjects were also asked for their food preference (sweet or salty taste).

During the first session, triangular taste detection thresholds for each solution were also determined after the GEP recordings.

#### 2.3.3. Triangular Taste Detection Threshold 

Gustatory thresholds were determined using a 3-alternative forced-choice procedure [20,21]. Participants were provided with successive sets of 3 samples; each set contained two control samples (Evian® water) and one stimulus sample (sucrose, aspartame or Stevia). Within each set, participants had to indicate which sample was different from the other 2. Sets were presented in ascending concentrations spaced by 0.25 log units. Sucrose (0.06090 to 1.08240 g), aspartame (0.00030 to 0.00540 g), or Stevia (0.00020 to 0.00360 g) was added to 100 mL of Evian® water (Table 1). The 6 concentrations of the 3 solutions used in the triangular taste detection threshold were noted as C1 to C6. If the subject did not recognize the sweet solution among the 3 samples, a set with the upper concentration was then presented to him/her. The procedure was completed when the subject correctly identified the stimulus sample at a given concentration three consecutive times. The concentration of each of the 3 sweet solutions was called the gustatory threshold of the considered solution for the participant. 

#### 2.3.4. GEP Recording and Data Analysis

Electroencephalographic (EEG) measurements were recorded according to the international 10-20 system using a conventional EEG recording system. Nine sites were recorded by surface electrodes defined by their scalp topography, as described elsewhere [18,19]: Centro-parietal electrode Pz, central electrode Cz, and frontal electrodes Fz, Fp1, Fp2. The electrodes were referenced against linked earlobes (ear clip electrodes enfolded by Ag, 10 mm diameter—SystemPLUS EVOLUTION). The ground electrode was placed on the forehead. Disposable cup electrodes enfolded by Ag-AgCl (6 mm diameter), with a long polyurethane cable (SystemPLUS EVOLUTION), were used. Electrodes were placed after first using a pumice paste and then a conductive and adhesive paste.

The EEG measurements were amplified, filtered, and digitized using Micromed software (SystemPLUS EVOLUTION, 2007 Micromed S.p.A), as follows: Time constant, 1 s; sampling frequency, 2048 Hz; 200 Hz low-pass filter; 0.4 Hz high-pass filter; and 50 Hz filter. GEPs were averaged after each recording session (average of 20 stimuli). No baseline correction was applied during averaging. 

GEP analysis was performed with the same software. Artifact corrections were applied during this analysis. GEP was defined by three peaks, as described in previous studies [18,19]: P1, the first positive peak; N1, the higher negative peak; and P2, the second positive peak. P1 latency (in ms), N1 latency (in ms), and P1N1 amplitude (in µV) of the GEPs were registered for each recorded electrode. The P1 latency was defined as the time interval between stimulus delivery and the potential positive peak P1. The N1 latency was defined as the time interval between the stimulus delivery and the potential’s negative peak. The amplitude of each response was calculated as the height between the first positive and the negative peaks (P1N1 amplitude). The positive peak corresponded to the peak pointing down, whereas the negative peak corresponded to the peak pointing up. The software first averaged the GEPs (n = 20) and then detected the peaks. A GEP recording in response to aspartame was rejected in one subject because of major artifacts. In 4 subjects, GEPs were not observed at the Fp1 and Fp2 electrodes in response to aspartame. In 2 of these 4 subjects, GEPs at the Fp1 and Fp2 were also not observed in response to Stevia.

At the end of the recordings with the same stimulus, an average of the responses of all subjects was performed: It was called the “grand average”. P1N1 amplitudes were minimized in the “grand average” compared to the statistical mean: In fact, there was a smoothing of the amplitude in the graph because GEP peaks of each subject did not have the same latency. The GEP recordings were analyzed by the same well-trained neurophysiologist and were processed with a standard and consistent method of EEG analysis, regardless of the quality and intensity of the taste solution and the hedonic value noted by the subject. The neurophysiologist was blinded regarding the taste solution applied. Due to constraints inherent to our software, pre-stimulus cerebral activity was not available. 

### 2.4. Statistical Analysis

Three comparisons (that is to say, comparisons of GEPs parameters in response to sucrose, aspartame, and Stevia stimulations in the same subjects) were planned with an alpha-risk of 0.05. According to previous results [19], a sample size of 18 achieved a 95% power to detect a mean of paired differences of P1 latencies of GEPs of 25 ms with an estimated standard deviation of differences of 19 ms and with a significance level (alpha) of 0.05 using a two-sided Wilcoxon test. Moreover, the number of participants was in the same range as recent studies with a comparable methodology [16,22,23].

Gender, a qualitative variable, was expressed in percentages. The P1 latency, N1 latency, P1N1 amplitude of each GEP, and the VAS results (for the hedonic value of the solutions) were expressed as the mean and standard deviation. These variables were compared between the 3 sweet solutions using the Friedman test for repeated measures on ranks. Post hoc analyses (Tukey’s tests) were also performed when the result was found to be significant, to isolate the group that was different from the others. The same analyses were performed to compare GEPs characteristics between genders. A Friedman test with 2 explaining factors (solutions and locations of electrodes) was also performed. Detection thresholds, which were discrete quantitative variables, were expressed as the median and 25%–75% interquartile range (IQR). Values were compared using the Friedman repeated measure test on ranks. Spearman correlation coefficients were calculated between VAS results (for the hedonic value of the solutions) and GEP parameters (P1 and N1 latencies and P1N1 amplitude). A p-value below 0.05 was considered statistically significant (preliminary results).

SAS 9.2 software (SAS Institute Inc., Cary, NC, USA) was used for all analyses.

## 3. Results

### 3.1. Saccharose, Aspartame, and Stevia Gustatory Thresholds

The absolute value of taste detection thresholds was different for the 3 sweet solutions (*p* < 0.001). In post hoc analyses, sucrose had a higher median detection threshold (0.10820; IQR: 0.08430 g/100 mL) than aspartame (0.00070; IQR: 0.00091 g/100 mL, *p* < 0.001) and Stevia (0.00064; IQR: 0.00028 g/100 mL, *p* < 0.001). Detection thresholds of aspartame and Stevia did not differ significantly from one another.

However, these solutions induced similar sweetening power. Hence, by comparing the range of gustatory threshold for the 3 sweet solutions (C1 to C6), no difference remained significant: The median ± IQR detection threshold was 3 ± 1 for sucrose, 3 ± 2 for aspartame, and 4 ± 1 for Stevia (*p* > 0.05), which are solutions with similar sweetening power.

### 3.2. Hedonic Value of the Solutions 

The hedonic values of each solution perceived during experimental sessions, and evaluated by VAS, were as follows: 5.3 ± 2.3 for sucrose, 5.6 ± 1.6 for aspartame, and 5.0 ± 2.0 for Stevia. No difference in VAS results was observed between the three hedonic values. 

No difference in VAS results was observed between genders.

### 3.3. GEP Recordings in Response to Sucrose, Aspartame, and Stevia Solutions, and Comparison of Their Parameters (P1 and N1 Latencies and P1N1 Amplitude) 

Two grand averages of GEP recordings were illustrated: The grand average of the Cz electrode in Figure 1 and the grand average of the Fp1 electrode in Figure 2.

P1 latencies were significantly different in response to the three taste stimuli at Pz (*p* < 0.001), Cz (*p* < 0.001), Fz (*p* < 0.01), Fp1 (*p* < 0.05), and Fp2 (*p* < 0.05). In post hoc analyses, P1 latencies were shorter when subjects were stimulated by the sucrose solution than when they were stimulated by the aspartame one at the Pz (*p* < 0.01), Cz (*p* < 0.05), and Fz electrodes (*p* < 0.05), and P1 latencies were shorter in response to the sucrose solution than to the Stevia one at the Pz (*p* < 0.001), Cz (*p* < 0.001), Fz (*p* < 0.001), Fp1 (*p* < 0.01) and Fp2 electrodes (*p* < 0.01), respectively. P1 latencies were also shorter when subjects were stimulated by the aspartame solution than the Stevia one at the Pz (*p* < 0.05), Cz (*p* < 0.05), Fz (*p* < 0.05), Fp1 (*p* < 0.05), and Fp2 (*p* < 0.05) electrodes. All the post hoc analyses are described in Figure 3.

In the test performed with two explaining factors (solutions and electrodes locations), P1 latencies were also different according to the solutions, and the electrodes. The results of P1 latencies were similar to those presented before. Concerning electrodes’ location, the P1 latencies in Fp1 and Fp2 electrodes were significantly longer than those recorded in Pz, Cz, and Fz (*p* < 0.001). 

N1 latency and P1N1 amplitudes (data showed in Table 2) did not differ between the three GEP recordings in response to sucrose, aspartame, and Stevia. No difference in GEP parameters was observed between genders. No significant correlation was noted between GEP parameters and the hedonic values of the three sweet solutions.

## 4. Discussion

In this study, we aimed to investigate GEPs in response to sweeteners (Stevia, a natural sweetener, and aspartame, an artificial sweetener), in comparison with GEPs obtained after stimulation by a sucrose solution. It has been observed that the P1 latency of the GEPs was longer when the subjects were stimulated by sweetener than by sucrose solution, regardless of the recording site. The P1 latencies were similar between sucrose and aspartame stimulation at the Fp1 and Fp2 electrodes only. Stevia was the solution that led to the longer P1 latencies compared with both sucrose and aspartame stimuli, with significant differences in all recording sites. It is important to understand the mechanisms of cerebral activation of natural sugar and sweeteners, which are responsible for a similar sweet taste but lead to different metabolic effects. Moreover, the use of sweeteners increases consumption and remains controversial in the literature data, especially for aspartame. In fact, aspartame could cause many problems such as neurological and behavioral disorders, which seem to be due to changes in cerebral concentrations of catecholamines. Steviol glycosides, natural sweeteners, represent a very interesting alternative. However, the physiological mechanisms underlying the sweet taste induced by Steviol glycosides and their cerebral activation remain unknown. 

The main result of this study was the observation of differences in P1 latencies of GEPs between the three sweet stimuli. It is one of the first studies that explored the gustatory pathway activated by sweeteners using GEPs, a higher time resolution technique than fMRI [17,24]. Indeed, contrary to cerebral activation in response to artificial sweeteners, which have been studied by fMRI [14,15,25], taste cortical responses after taste receptor stimulation by Stevia solutions had never been explored. Several hypotheses should be considered to explain the results of the present study. 

First, differences observed between GEP latencies in response to sucrose and sweeteners could be explained by the different taste receptors involved in the detection of these substances. Certain authors argued that the T1R2-T1R3 heterodimer taste receptor of sugar responds to all classes of sweet tastants, including natural sugars and artificial sweeteners [26,27], whereas other works indicated that T1R3 could be the only sub-unit that is able to detect sweeteners [28]. Indeed, it has recently been observed that stevioside and rebaudioside A, components of Stevia, interact with the T2R4 receptor, which is a bitter taste receptor [29,30]. However, the hypothesis of differences in taste receptors is not satisfying enough, because certain authors demonstrated that GEPs obtained with the same experimental method had similar latencies in response to sweet, bitter, umami, salty, and sour tastes [17]. 

Second, the differences observed in the delay of GEPs (P1 latency) after sweet taste stimulus between sucrose and sweeteners could be explained by the differential activation of cerebral areas in response to these substances. fMRI studies have established that the human brain could distinguish a sweet nutritive taste (for example, sucrose) from a non-nutritive one (for example, aspartame or stevia) [31]. Several studies suggested that natural sugar was better able to activate the hedonic pathway of food intake (anterior insula, frontal operculum, striatum, anterior cingulate cortex, ventral tegmental area, and amygdala) than artificial sweeteners [14,15,25,32,33,34]. A recent study using electroencephalography (EEG) and Gustatory Evoked Potentials (GEPs) methods [16] also observed discrepancies in brain activation after stimulation by different sweet solutions (sucrose and aspartame).

Third, differences in cortical gustatory responses observed between sucrose, aspartame, and Stevia using GEPs could be related to differential glucose blood levels. In fact, cerebral gustatory pathway activation is modulated by glucose blood levels [35,36,37]. Sucrose ingestion increases glucose blood level, whereas aspartame or Stevia consumption does not. Therefore, we could expect that sucrose leads to greater cerebral activation in the gustatory pathway than aspartame or Stevia solutions, through activation of brain glucose sensor neurons in the hypothalamus and brainstem [35,36,37]. However, it is difficult to understand how this hypothesis could explain a phenomenon that occurs so rapidly after sweet ingestion (differences from 150.9 ms to 203.8 ms at the Cz electrodes, for instance), especially considering that the amount of sucrose delivered was very small (20 stimuli of 1 second with a 200 mL/h flow rate of a 10 g sucrose per 100 mL of water solution). 

Fourth, longer GEP latencies after aspartame and particularly Stevia stimulation could be explained by the fact that the subjects were not regular consumers of these sweeteners. In fact, the sweet taste of natural sugar is considered as an innate human preference, which is apparent in infancy, childhood, and even during fetal development [38,39,40,41,42]. The sweet conditioning could lead to learning-related changes in the neural gustatory processing of the tastes themselves, such as an increase in synaptic connections due to neural plasticity [43,44,45,46,47]. An increase in synaptic connections linked to natural sugar taste recognition, as opposed to the taste of sweeteners, could explain the differences observed in GEP latencies: The larger the number of synaptic connections, the faster the cerebral response. A similar process has been demonstrated in habitual users of artificial sweeteners who have greater activation of reward-related brain regions, such as the orbitofrontal cortex, in response to saccharin taste [48].

Another question remains unresolved. It is unclear why subjects had longer GEP latencies with Stevia, a natural sweetener, than with aspartame, an artificial sweetener. Is it due to the bitterness of Stevia, which is more intense than aspartame, and which activates aversive gustatory pathways, even if subjects are not conscious of distinguishing it [29,30,49]? Considering Stevia as an aversive taste could explain the lesser activation of the gustatory cortex leading to longer GEP latencies, as was demonstrated after food intake [18]. 

The experimental design used in this study had a few limitations that were exposed in detail elsewhere [18,19]. Briefly, GEPs have a poor spatial resolution, which contrasts with an excellent time resolution [17]. The insula, a major part of the primary gustatory cortex, is not easy to record using GEPs because it is deep in the brain. We cannot totally exclude the possibility that somatosensory cortical responses may have contributed slightly to the GEPs [50,51], though the high-time resolution technique we used was able to distinguish between gustatory and somatosensory responses (different latencies). We cannot totally exclude the possibility that the gustatory cortex was also activated by water [52,53,54,55]. Although uniform stimulation of a large lingual surface was applied to activate the same oral receptive fields in each subject, we cannot exclude small movements of the tube in the mouth. We cannot exclude the possibility of a small delay (a few milliseconds) between the delivery of the taste solution and activation of the taste receptors. 

In addition to a few limits inherent to the experimental design, which were exposed elsewhere [18,19], a limitation was specific to the present study investigating the effects of sweeteners on the gustatory pathway. Artificial and natural sweeteners can be perceived with a bitter aftertaste in the mouth. The bitterness is known to be more intense for Stevia than for aspartame solutions and tends to increase with high concentrations of sweeteners [29,49,56]. Perceptual interactions of sweet and bitter tastes have been attributed to peripheral mechanisms occurring at the receptor level [57] or to central processing due to neural inhibition [58]. However, none of the participants in the present study detected a bitter aftertaste when they received sweetener stimuli because moderate concentrations of each stimulus were used as a precaution. 

The method used in this study for taste stimulation and GEP recordings was verified in previous studies and demonstrated valid results when subjects were stimulated by sucrose solutions with different intensities and hedonic values [19] and according to food intake [18]. Additionally, the GEPs obtained after stimulation by the sucrose solutions in this study had similar parameters (P1 and N1 latencies, P1N1 amplitude) as those recorded in the previous studies with the same experimental method [18,19,59,60]. 

## 5. Conclusions

In conclusion, this study identifies the difference of cerebral activation in response to sucrose and sweeteners (aspartame and Stevia) using GEPs: P1 latency of GEPs was significantly shorter after sucrose stimulus than after aspartame or Stevia stimuli. Many hypotheses could be generated. Besides differences of taste receptors and cerebral areas activated by these substances, neural plasticity, and changes in synaptic connections related to sweet innate preference and sweet conditioning, could be the best hypothesis to explain the differences in cerebral gustatory processing after sucrose and sweeteners activation. In fact, the sweet conditioning during infancy and childhood could lead to learning-related changes in the neural gustatory processing of the tastes themselves, such as an increase in synaptic connections due to neural plasticity. This increase in synaptic connections linked to natural sugar taste recognition as opposed to the taste of sweeteners could explain the differences observed in GEP latencies: The larger the number of synaptic connections, the faster the cerebral response.

## Figures and Tables

**Figure 1 nutrients-12-00322-f001:**
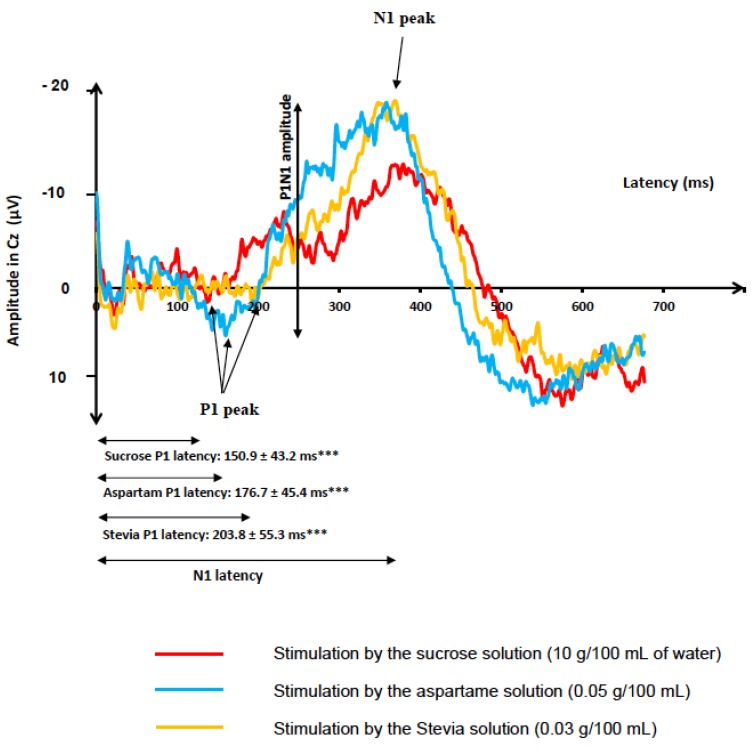
Grand average (average of the responses of all subjects) of gustatory-evoked potentials (GEPs): Recordings of GEPs in response to the three sweet solutions (sucrose, aspartame, and Stevia) at the Cz electrodes, for all the 20 participants. GEP was defined by three peaks: P1, the first positive peak; N1, the higher negative peak; and P2, the second positive peak. The start of the taste stimulation was at 0 ms. P1 latencies were significantly different between the three taste stimuli in GEPs recorded at Cz. *** indicates *p* <0.001 for the global statistical model comparing P1 latencies of GEPs in Cz between the three solutions (Friedman test on ranks).

**Figure 2 nutrients-12-00322-f002:**
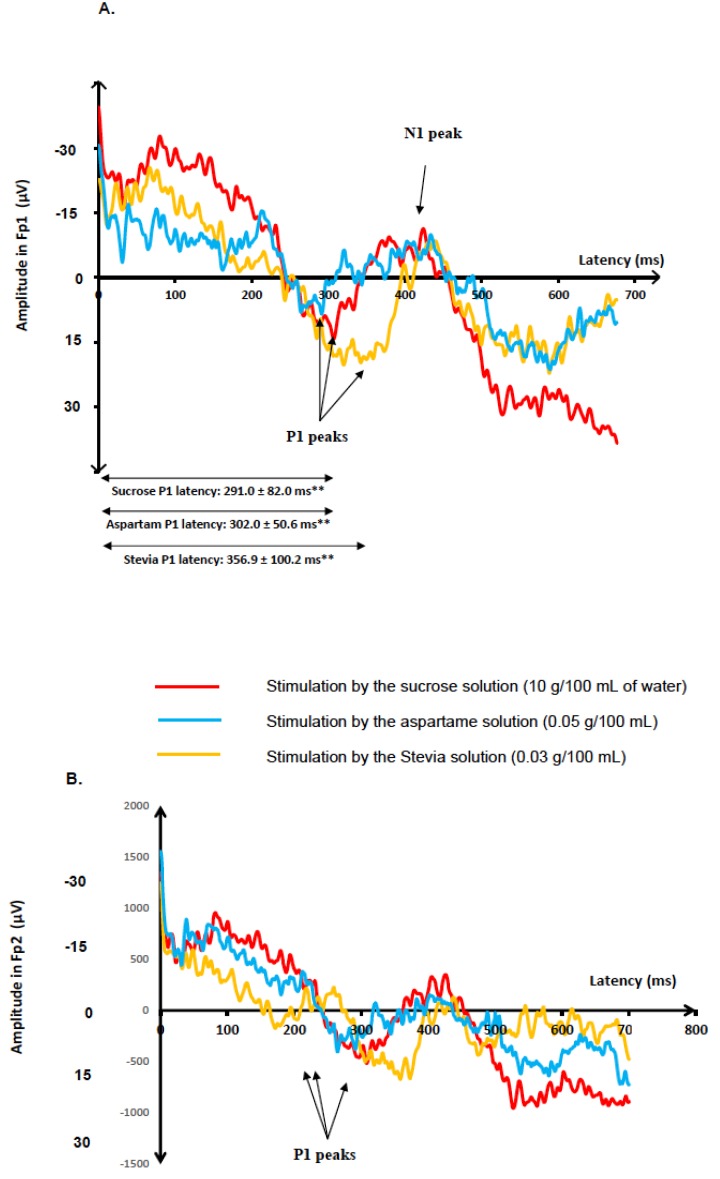
Grand average (average of the responses of all subjects) of gustatory-evoked potentials (GEPs): Recordings of GEPs in response to the three sweet solutions (sucrose, aspartame, and Stevia) at the Fp1 (Figure A) and Fp2 (Figure B) electrodes, for the 20 participants. GEP was defined by three peaks: P1, the first positive peak; N1, the higher negative peak; and P2, the second positive peak. The start of the taste stimulation was at 0 ms. P1 latencies were significantly different between the three taste stimuli in GEPs recorded at Fp1. ** indicates *p* < 0.01 for the global statistical model comparing P1 latencies of GEPs in Fp1 between the three solutions (Friedman test on ranks).

**Figure 3 nutrients-12-00322-f003:**
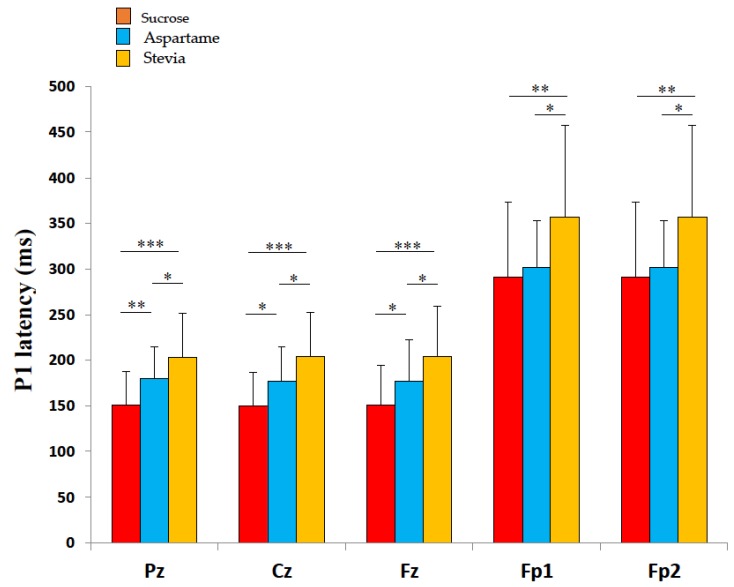
Mean (± SD) P1 latency of the gustatory evoked potentials (GEPs) of the 20 subjects, in response to the three sweet solutions: Sucrose 10 g/100 mL, aspartame 0.05 g/100 mL, and Stevia 0.03 g/100 mL of water. P1 latencies of the GEPs differed significantly between the three sweet taste stimuli, for all the recording sites (Pz, Cz, Fz, Fp1, and Fp2). Significant differences in comparisons between the taste solutions obtained with post hoc analyses are expressed as follows: **p* < 0.05; ***p* < 0.01, ****p* < 0.001.

**Table 1 nutrients-12-00322-t001:** Concentrations of sucrose, aspartame, and Stevia solutions used to evaluate subjects’ taste detection thresholds.

Sucrose	Concentration g/100 mL	Aspartame	Concentration g/100 mL	Stevia	Concentration g/100 mL
C1	0.06090	C1	0.00030	C1	0.00020
C2	0.10820	C2	0.00054	C2	0.00036
C3	0.19250	C3	0.00096	C3	0.00064
C4	0.34230	C4	0.00170	C4	0.00110
C5	0.60870	C5	0.00300	C5	0.00200
C6	1.08240	C6	0.00540	C6	0.00360

**Table 2 nutrients-12-00322-t002:** Mean ± SD of the GEPs parameters (P1 latency, N1 latency, and P1N1 amplitude) recorded in response to the three sweet solutions on the Pz, Cz, Fz, Fp1, and Fp2 electrodes, in the 20 participants.

GEP Characteristics		Sucrose	Aspartame	Stevia
**P1 latency (ms)**	**Pz**	150.5 ± 36.8	179.3 ± 35.5	203.4 ± 48.2
	**Cz**	149.8 ± 21.0	176.4 ± 38.1	204.2 ± 48.0
	**Fz**	150.9 ± 43.2	176.7 ± 45.4	203.7 ± 55.3
	**Fp1**	283.4 ± 95.3	334.6 ± 70.9	356.9 ± 100.2
	**Fp2**	282.9 ± 97.1	335.0 ± 71.2	356.2 ± 99.9
**N1 latency (ms)**	**Pz**	347.0 ± 53.7	341.2 ± 94.4	350.8 ± 64.1
	**Cz**	350.2 ± 46.3	338.6 ± 94.1	350.6 ± 64.0
	**Fz**	350.7 ± 45.6	338.2 ± 94.5	348.1 ± 64.3
	**Fp1**	417.5 ± 58.1	424.9 ± 97.4	429.3 ± 67.6
	**Fp2**	418.11 ± 61.4	422.0 ± 101.1	426.1 ± 70.9
**P1N1 amplitude (µV)**	**Pz**	16.0 ± 7.1	16.6 ± 9.2	15.8 ± 7.4
	**Cz**	15.2 ± 6.6	16.8 ± 10.5	13.8 ± 6.3
	**Fz**	14.5 ± 6.9	14.1 ± 8.5	13.4 ± 6.0
	**Fp1**	18.6 ± 13.9	19.0 ± 13.8	24.4 ± 15.2
	**Fp2**	18.4 ± 16.0	19.9 ± 15.4	24.6 ± 15.4

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
