# Peer review of "Differential Cerebral Gustatory Responses to Sucrose, Aspartame, and Stevia Using Gustatory Evoked Potentials in Humans"

_nutrients, 2020, doi:10.3390/nu12020322_

Round 1
Reviewer 1 Report
The authors studied cerebral gustatory responses to sucrose, aspartame and stevia using gustatory evoked potentials in humans. They identified the difference of cerebral activation in response to sucrose and sweeteners. This study is interesting but there are some issues in the figures listed below.
1) Figure 1: the unit on y axis should be μV; the number "-1000" is still on y axis (looks like there are two y axes overlapped); the two-way arrow and the text are overlapped on the bottom of the figure.
2) Figure 2: the two-way arrow and the text are overlapped on the bottom of the figure.
Author Response
The authors studied cerebral gustatory responses to sucrose, aspartame and stevia using gustatory evoked potentials in humans. They identified the difference of cerebral activation in response to sucrose and sweeteners. This study is interesting but there are some issues in the figures listed below.
1) Figure 1: the unit on y axis should be μV; the number "-1000" is still on y axis (looks like there are two y axes overlapped); the two-way arrow and the text are overlapped on the bottom of the figure.
2) Figure 2: the two-way arrow and the text are overlapped on the bottom of the figure.
In fact, errors in the figures correspond to problems during the formatting of the figures in the article. We modified the configuration of our figures to avoid this problem.
Reviewer 2 Report
This study identifies the difference of cerebral activation in response to sucrose and 427 sweeteners (aspartame and Stevia) using GEPs. The design of the study is very in detail and the results are reasonable based on evidence. Revision to the following points are required.
First, author mentioned “no study has compared cerebral activation 65 between sugar and natural sweeteners such as Stevia”. I know that a few similar research exists on the this topic. We recommend that authors add relevant research.
Second, the hypotheses presented in the conclusions need to be presented more elaborately and diversely, and if there is any previous research related to these hypotheses, authors need discuss implication and contribution of authors’ findings to previous findings.
Third, in conclusion, it is necessary to mention the methodological limitations of this study. In particular, it is necessary to present the limitations of the experimental design adopted in this study.
Author Response
This study identifies the difference of cerebral activation in response to sucrose and 427 sweeteners (aspartame and Stevia) using GEPs. The design of the study is very in detail and the results are reasonable based on evidence. Revision to the following points are required.
First, author mentioned “no study has compared cerebral activation 65 between sugar and natural sweeteners such as Stevia”. I know that a few similar researches exist on this topic. We recommend that authors add relevant research.
As suggested, we added two references concerning Stevia and reward system on the one hand, and Stevia and appetite on the other hand (introduction section) (lines 65 to 69).
Second, the hypotheses presented in the conclusions need to be presented more elaborately and diversely, and if there is any previous research related to these hypotheses, authors need discuss implication and contribution of authors’ findings to previous findings.
As suggested, we added more details concerning our hypotheses in the conclusion section (lines 449 to 453).
Third, in conclusion, it is necessary to mention the methodological limitations of this study. In particular, it is necessary to present the limitations of the experimental design adopted in this study.
We added limitations of the experimental design in the limitation section, just before the conclusion (lines 413 to 424).
This manuscript is a resubmission of an earlier submission. The following is a list of the peer review reports and author responses from that submission.
Round 1
Reviewer 1 Report
This research studied cerebral gustatory responses to sucrose, aspartame and stevia using gustatory evoked potentials in humans. They identified the difference of cerebral activation in response to sucrose and sweeteners. Suggest a few revisions list below:
1) Figure 1 and figure 2 have some white squares on it. In figure 1, the number "-1000" on Y axis is confusing.
2) In "Conclusion", "this study identifies the modifications of cerebral activation in response to sucrose and sweeteners" should be "this study identifies the difference of cerebral activation in response to sucrose and sweeteners". The observed activation is the response to different stimulations, but it is not modified.
Author Response
This research studied cerebral gustatory responses to sucrose, aspartame and stevia using gustatory evoked potentials in humans. They identified the difference of cerebral activation in response to sucrose and sweeteners. Suggest a few revisions list below:
1) Figure 1 and figure 2 have some white squares on it. In figure 1, the number "-1000" on Y axis is confusing.
We thank reviewer 1 for his advice. As suggested, we modified and corrected the figure 1 in the manuscript. “-1000” is an error due to the formatting of the figures between our files and the revue.
2) In "Conclusion", "this study identifies the modifications of cerebral activation in response to sucrose and sweeteners" should be "this study identifies the difference of cerebral activation in response to sucrose and sweeteners". The observed activation is the response to different stimulations, but it is not modified.
As suggested, we clarified and modified the sentence in the conclusion in the manuscript (line 449).
Reviewer 2 Report
This a well-written paper with a clearly described experimental setup and with a reasonable introduction and discussion.
Only one remark: please explore the difference between "Stevia" and "steviol glycosides".
Author Response
This a well-written paper with a clearly described experimental setup and with a reasonable introduction and discussion.
Only one remark: please explore the difference between "Stevia" and "steviol glycosides".
We thank reviewer 2 for his advice.
Stevia is the south American plant and Steviol glycosides are the chemical compounds responsible for the sweet taste of the leaves of the plant Stevia. The main steviol glycosides are rebaudioside A and stevioside, used as artificial sweeteners and 100–300 times sweeter than sucrose (Samuel P et al, J Nutr, 2018).
We added some information in the first paragraph of introduction.
Reviewer 3 Report
This studies analyzed differential cerebral gustatory responses to sucrose, 2 aspartame and Stevia using gustatory evoked 3 potentials in humans. The experimental design is elaborately designed and the results are very interesting. However, the following revisions are required.
Literature review of studies related to this study should be done in advance before experiment. Since there are many previous studies on this subject, it is necessary to emphasize the differentiation of this study after reviewing the existing literature.
Additional information on the participants in experiment should be provided. Currently, since only basic information is provided, it is difficult to judge the state of the experimental group.
Author Response
This studies analyzed differential cerebral gustatory responses to sucrose, 2 aspartame and Stevia using gustatory evoked 3 potentials in humans. The experimental design is elaborately designed and the results are very interesting. However, the following revisions are required.
Literature review of studies related to this study should be done in advance before experiment. Since there are many previous studies on this subject, it is necessary to emphasize the differentiation of this study after reviewing the existing literature.
We thank reviewer 3 for his advice. We complete the introduction with a rationale of this study about effects of sweeteners (lines 56-63):
“The use of sweeteners is increasingly controversial in the literature, especially for aspartame (Choudhary AK et al, Nutr Rev, 2017; Magnuson BA et al, Nutr Rev, 2018). Among the disorders caused by the consumption of aspartame, neurological and behavioral disorders have been reported, including headache, insomnia and seizures. These disorders may be due to changes in regional brain concentrations of catecholamines induced by the metabolites of aspartame (Humphries P et al, Eur J Clin Nut, 2007). The steviol glycosides from Stevia represent a very interesting alternative. However, the physiological mechanisms of action of these glycosides are still unknown, particularly concerning cerebral activation induced by this natural sweetener (Samuel P et al, J Nutr, 2018).”
Additional information on the participants in experiment should be provided. Currently, since only basic information is provided, it is difficult to judge the state of the experimental group.
As suggested, we added information about participants: “All the participants practiced regular physical activity (about 30 minutes 3 times per week). They did not have special diet (vegan, vegetarian…) and their meal was balanced. They consumed rarely sweeteners or prepared foods which contain sweetenerrs (< 1 time per month). The participants drank about 1 to 3 coffee per day. They drank alcohol occasionally (1-3 glasses per week), without abnormality according to WHO (World Health Organization) guidelines.” (lines 106-111). (https://www.who.int/substance_abuse/publications/alcohol/en/).
Reviewer 4 Report
The authors examined the impacts of aspartame and stevia on cerebral activation, as compared to sucrose.
This is an interesting study. However, there are some concerns pertaining to the study rationale, sample size calculations and statistical inference which limit enthusiasm.
ABSTRACT
MAJOR COMPULSORY REVISIONS:
Unclear why cerebral activation of interest. What message does this study have?MINOR COMMENTS:
The sentences are too long about the results. For this study, location of EEG electrodes is not important. Please remove that information because of complicating. In terms of the p-values, please indicate where is significantly different from what, clearly. And please indicate effect sizes. The authors need to define EEG and VAS.
INTRODUCTION
MAJOR COMPULSORY REVISIONS:
This reviewer does not believe an adequate rationale has been provided for the study. How important is this study? And why? Particularly, EEG-measured cerebral activation is returned to baseline 1 sec after stimulation, please explain that physiological observation meanings, especially about P1 latency, N1 latency, and P1N1 amplitude. Please explain the difference between benefits of gustatory stimulation and glucose ingestion effects. Why do the authors have to examine gustatory stimulation impacts, but not ingestion?MINOR COMPULSORY REVISIONS
What is “peripherally mediated” meaning? (L50) L66-69; this sentence is “hypothesis of hypothesis”. Please rephrase. L69-70; Where is “this purpose”? Probably, hypothesis? Please rephrase (e.g., To test these hypotheses)
METHODS
MAJOR COMPULSORY REVISIONS:
Participants: How about alcohol, caffeine, and fitness level? Please show about the age difference between men and women because the authors have mentioned the sex difference in the Results section. Protocol: How did the authors get stimulation signal? Did the authors use the triggers? The authors indicate only differences in the P1 latency, so this is very important. Protocol: Why did the authors use 1 min 20 sec stimulation and 1 min interval? Protocol: L168 (average of 20 stimuli); Please indicate SD on each condition because of important factor. Condition: Why did the authors perform this experiment 2-4 h after the participants had a lunch? Even though blood glucose is returned to baseline level, brain behavior and physiology may be affected by post-prandial blood glucose elevation, hyperinsulinemia, and hyperlipidemia, etc at least 2 h after meal consumption. Do the authors have control data (e.g., baseline measurement in response to same stimulation)? Is there day-to-day variations in the baseline measurement of GEPs? Sample size: Did the authors perform power calculation before experiment? Statistics: Are there normal distribution? The authors must confirm the normal distribution before performing ANOVA. In addition, did the authors check sphericity assumption? If the sphericity assumption was not met, corrections should be used such as Greenhouse-Geisser. Statistics: Why do the authors use ONE-way ANOVA? How about the location (Pz, Cz, Fz, Fp1, and Fp2) effects? Statistics: The authors should calculate the effect size because the magnitude of differences can not be determine using only P-values.
MINOR REVISIONS
Participants: L120; what is 30 different subjects meaning? 20 subjects took part in this study, right? Protocol: Double or Single blind? Table 1: Pleas explain C1 to C6. The authors need to define ANOVA.
RESULTS
MAJOR COMPULSORY REVISIONS:
How important is N1 latency and P1N1 amplitude, as compared to P1 latency? How about the P1N1 latency? Is P1N1 latency not acceptable to determine GEPs? Probably, whereas change in P1N1 latency would be opposite to P1 latency because of no changing N1 latency, the authors examined P1 latency, N1 latency, and P1N1 amplitude. Why did the authors determine the P1N1 amplitude, whereas why not determine the P1N1 latency? Which are important; P1 or N1 or P1N1 difference? It looks like contradiction. L214 and 283-284; How did the authors calculate the statistical sex differences? Figure 1 and 2; Please reformat the figures (e.g., y axis is unclear, what is -1000? etc). And, the authors need to present all location (Why did the authors only present C2 and Fp1?). In addition, P2 peak should be removed because it is not necessary for this study. Figure 1 and 2; What is “* indicates P < 0.00X for the global statistical model (one-way ANOVA for repeated measures)” meaning?MINOR REVISIONS
L214; Insert 3.3? Figure 3; Please present N1 latency and P1N1 amplitude data at least in the Table or Supplemental figure.
DISCUSSION
MAJOR COMPULSORY REVISIONS:
In the discussion, please explain the novelty, significance, and usefulness of this study. What are the impact and importance of this study? How do this study contribute to promote science as a new research article? Second and Third paragraphs (L294-310) should be moved to limitation section, which should be located before conclusion. And the authors should remove “which strengthens the reliability of the present results” because of speculative interpretation. Sixth paragraph (L294-310); the authors should not mention EEG location. If the authors would like to explain the differential cerebral activation, the P1 latency has to have interaction using 2-way (condition * location) ANOVA (and then if P1 latency indicate same results following post-hoc comparisons, the authors can mention EEG location).MINOR REVISIONS
L320; Please relocate the reference (or please remove period).
CONCLUSION
The authors should only describe a new finding.
Author Response
The authors examined the impacts of aspartame and stevia on cerebral activation, as compared to sucrose.
This is an interesting study. However, there are some concerns pertaining to the study rationale, sample size calculations and statistical inference which limit enthusiasm.
ABSTRACT
MAJOR COMPULSORY REVISIONS:
Unclear why cerebral activation of interest. What message does this study have?
We briefly specified the rationale of the study (Lines 24-25).
We explained the main message of our study at the end of the abstract (Lines 36-41).
MINOR COMMENTS:
The sentences are too long about the results. For this study, location of EEG electrodes is not important. Please remove that information because of complicating. In terms of the p-values, please indicate where is significantly different from what, clearly. And please indicate effect sizes. The authors need to define EEG and VAS.
As indicated, we shortened the sentences of the results by removing the location of EEG electrodes (Lines 29-35).
We explained EEG (Line 29) and VAS (Line 30).
Effect sizes could be very interesting, but they are often used for studies with wide extent, such as meta-analyses or power study. That’s why we did not calculated effect sizes in our study, according to our colleague who is specialized in statistics.
INTRODUCTION
MAJOR COMPULSORY REVISIONS:
This reviewer does not believe an adequate rationale has been provided for the study. How important is this study? And why? Particularly, EEG-measured cerebral activation is returned to baseline 1 sec after stimulation, please explain that physiological observation meanings, especially about P1 latency, N1 latency, and P1N1 amplitude. Please explain the difference between benefits of gustatory stimulation and glucose ingestion effects. Why do the authors have to examine gustatory stimulation impacts, but not ingestion?
We complete the introduction with a rationale of this study about effects of sweeteners (lines 56-63):
“The use of sweeteners is increasingly controversial in the literature, especially for aspartame (Choudhary AK et al, Nutr Rev, 2017; Magnuson BA et al, Nutr Rev, 2018). Among the disorders caused by the consumption of aspartame, neurological and behavioral disorders have been reported, including headache, insomnia and seizures. These disorders may be due to changes in regional brain concentrations of catecholamines induced by the metabolites of aspartame (Humphries P et al, Eur J Clin Nut, 2007). The steviol glycosides from Stevia represent an very interesting alternative. However, the physiological mechanisms of action of these glycosides are still unknown, particularly concerning cerebral activation induced by this natural sweetener (Samuel P et al, J Nutr, 2018).”
As suggested, we described briefly the GEP recordings, as described in our previous reports (Jacquin-Piques et al, Front Neurosci 2016 - Jacquin-Piques et al, Chem Senses 2016 – Mouillot et al, J Lipid Res 2019 – Mouillot et al, Chem Senses 2019) à Lines 79-86.
It would be very interesting to study the effects of sweet ingestion on cerebral activation. However, sweet ingestion involved activation of the gustatory receptors, activation of the gustatory afferent pathway until the cerebral gustatory areas and the relationship between gut and brain. Gustatory stimulation without ingestion involved the same steps except the relationship between gut and brain which corresponds to complex links. So, it is important to first understand how are activations of the gustatory pathway from taste receptors to cerebral gustative areas between the three studied sweet solutions. Then, we could conduct a new study with ingestion of sucrose, aspartame and Stevia to observe the effects of ingestion on the cerebral activation ; we considered this suggestion for another study.
MINOR COMPULSORY REVISIONS
What is “peripherally mediated” meaning? (L50) L66-69; this sentence is “hypothesis of hypothesis”. Please rephrase. L69-70; Where is “this purpose”? Probably, hypothesis? Please rephrase (e.g., To test these hypotheses)
“peripherally mediated” means mediated by the peripheral nervous system (contrary ti the central nervous system). We removed this term for more clarity (Line 64).
As suggested, we rephrased the hypothesis of the study (Lines 93 to 95).
METHODS
MAJOR COMPULSORY REVISIONS:
Participants: How about alcohol, caffeine, and fitness level? Please show about the age difference between men and women because the authors have mentioned the sex difference in the Results section. Protocol: How did the authors get stimulation signal? Did the authors use the triggers? The authors indicate only differences in the P1 latency, so this is very important. Protocol: Why did the authors use 1 min 20 sec stimulation and 1 min interval? Protocol: L168 (average of 20 stimuli); Please indicate SD on each condition because of important factor. Condition: Why did the authors perform this experiment 2-4 h after the participants had a lunch? Even though blood glucose is returned to baseline level, brain behavior and physiology may be affected by post-prandial blood glucose elevation, hyperinsulinemia, and hyperlipidemia, etc at least 2 h after meal consumption. Do the authors have control data (e.g., baseline measurement in response to same stimulation)? Is there day-to-day variations in the baseline measurement of GEPs? Sample size: Did the authors perform power calculation before experiment? Statistics: Are there normal distribution? The authors must confirm the normal distribution before performing ANOVA. In addition, did the authors check sphericity assumption? If the sphericity assumption was not met, corrections should be used such as Greenhouse-Geisser. Statistics: Why do the authors use ONE-way ANOVA? How about the location (Pz, Cz, Fz, Fp1, and Fp2) effects? Statistics: The authors should calculate the effect size because the magnitude of differences can not be determine using only P-values.
As suggested, we added information about participants: “All the participants practiced regular physical activity (about 30 minutes 3 times per week). They did not have special diet (vegan, vegetarian…) and their meal was balanced. They consumed rarely sweeteners or prepared foods which contain sweetenerrs (< 1 time per month). The participants drank about 1 to 3 coffee per day. They drank alcohol occasionally (1-3 glasses per week), without abnormality according to WHO (World Health Organization) guidelines.” (lines 106-111). (https://www.who.int/substance_abuse/publications/alcohol/en/).
As suggested, we added information about mean age and BMI for men and women (Lines 102-103).
To get stimulation signal, the recording system get the trigger of the stimulation which is send by an electronic device, as explained in the paragraph line 117 to 126.
We used 1 sec stimulation with 1 min interval to be sure not to stimulate in the refractory period of the receptors.
We have not SD for the average of the 20 GEP recordings, because the averaging is directly performed by the EEG computer software. This averaging is used to individualize the GEP from the background noise. It is the protocol used for all the recording of the evoked potentials in clinical and research practice. We have only SD for the “grand average”.
We performed gustatory stimulations 2-4 h after the end of lunch because we demonstrated in a previous study (Jacquin-Piques A et al, Front Neurosci 2016) that GEPs were modified 30-60 minutes after end of eating, probably linked to negative alliesthesia. Moreover, all the subjects were evaluated in the same physiological state, which allowed us to compare the results between subjects.
There is no day-to-day significant variations, as demonstrated in a previous study (Jacquin-Piques A et al, Chem Senses 2016 – Mouillot T et al, Chem Senses 2019); all the recordings were performed 2-4h after eating.
We performed a sample size calculation before the beginning of the study (Lines 223 to 229).
The distribution are not normal. So, we used a Friedman test for repeated measures on ranks for all the analyses. The significant results were similar (lines 307-309 in results). As a consequence, the sphericity, which is a condition for using ANOVA, is not necessary.
As suggested, we made new comparisons using 2 factors: solutions (sucrose, aspartame, stevia) and electrodes location. The two factors were significant (p<0,001). We presented these results in the article, in addition with the first results initially described. In fact, comparisons between electrodes which record different cerebral areas could be surprising in clinical practice. So, we preferred keeping the two analyses (Lines 236-237 for methods; lines 319-321 for results).
Effect sizes could be very interesting, but they are often used for studies with wide extent, such as meta-analyses or power study. That’s why we did not calculated effect sizes in our study, according to our colleague who is specialized in statistics.
MINOR REVISIONS
Participants: L120; what is 30 different subjects meaning? 20 subjects took part in this study, right? Protocol: Double or Single blind? Table 1: Pleas explain C1 to C6. The authors need to define ANOVA.
Right, we included 20 subjects. We corrected this data line 151.
It is a double blind protocol (we added this information line 153).
C1 to C6 corresponded to the 6 concentrations of the solutions used in triangular taste detection threshold (Lines 178-179).
As suggested before, we removed ANOVA.
RESULTS
MAJOR COMPULSORY REVISIONS:
How important is N1 latency and P1N1 amplitude, as compared to P1 latency? How about the P1N1 latency? Is P1N1 latency not acceptable to determine GEPs? Probably, whereas change in P1N1 latency would be opposite to P1 latency because of no changing N1 latency, the authors examined P1 latency, N1 latency, and P1N1 amplitude. Why did the authors determine the P1N1 amplitude, whereas why not determine the P1N1 latency? Which are important; P1 or N1 or P1N1 difference? It looks like contradiction. L214 and 283-284; How did the authors calculate the statistical sex differences? Figure 1 and 2; Please reformat the figures (e.g., y axis is unclear, what is -1000? etc). And, the authors need to present all location (Why did the authors only present C2 and Fp1?). In addition, P2 peak should be removed because it is not necessary for this study. Figure 1 and 2; What is “* indicates P < 0.00X for the global statistical model (one-way ANOVA for repeated measures)” meaning?
According to literature data, parameters useful to describe GEPs are P1 latency (the beginning of the cerebral response), N1 latency (which corresponds to latency of the GEP peak, the maximum of the cerebral response) and GEP amplitude, also called in our study P1N1 amplitude. P1N1 amplitude corresponds to the height of the GEP (height in µV between the positive P1 peak and the negative N1 peak), and it is probably linked to the maximal number of depolarized neurons. We reformulated the definition in the methods to make it clearer (line 208). These three parameters are important to describe GEPs (see Ohla et al, 2012 for review and the papers included in this review such as Hummel et al, 2010).
P1N1 latency (difference between N1 latency and P1 latency), as you suggested, is not a parameter used to described evoked potentials, in research or in clinical practice. Whereas P1 latency, N1 latency and P1N1 are a clinical translation, P1N1 latency doesn’t correspond to a physiological mechanism and so, it is useless to understand mechanism of GEPs. That is why we did not mention P1N1 latency (and it was not for the reason you noted in your report).
We calculated the statistical sex difference with Friedman test on ranks. We added this information in the methods paragraph (line 236).
As suggested, we reformatted the figures. “-1000” is an error due to the formatting of the figures between our files and the revue. We added figures for all the electrodes location which were analyzed. We did not present all the figures because Fz and Pz were similar to Cz on the one hand, and Fp2 was similar to Fp1 on the other hand.
As you suggested, we removed P2 peak.
Figure 1 and 2: *** or ** corresponds to the statistical results of the global model used to compare GEPs parameters between the three sweet solutions.
MINOR REVISIONS
L214; Insert 3.3? Figure 3; Please present N1 latency and P1N1 amplitude data at least in the Table or Supplemental figure.
Yes, it is right, we added a 3.3 (Line 260).
We added N1 latency and P1N1 amplitude data in Table 2.
DISCUSSION
MAJOR COMPULSORY REVISIONS:
In the discussion, please explain the novelty, significance, and usefulness of this study. What are the impact and importance of this study? How do this study contribute to promote science as a new research article? Second and Third paragraphs (L294-310) should be moved to limitation section, which should be located before conclusion. And the authors should remove “which strengthens the reliability of the present results” because of speculative interpretation. Sixth paragraph (L294-310); the authors should not mention EEG location. If the authors would like to explain the differential cerebral activation, the P1 latency has to have interaction using 2-way (condition * location) ANOVA (and then if P1 latency indicate same results following post-hoc comparisons, the authors can mention EEG location).
We explained better why these results could be important for the sweeteners consumption, which is the novelty and the usefulness of this study (lines 345 - 352)
As suggested, we moved second and third paragraphs to limitation section before conclusion and we remove the last sentence: “which strengthens the reliability of the present results”.
Sorry, we don’t understand the last remark (because we did not mention EEG location in this paragraph). Did the reviewer want to say that we should remove the literature data concerning cerebral areas potentially activated by natural sugar and sweeteners?
MINOR REVISIONS
L320; Please relocate the reference (or please remove period).
Ok, we removed it.
CONCLUSION
The authors should only describe a new finding.
Sorry, we don’t also understand this remark about conclusion; we mentioned the only result of our study.